# Repurposing Disulfiram for Targeting of Glioblastoma Stem Cells: An In Vitro Study

**DOI:** 10.3390/biom11111561

**Published:** 2021-10-21

**Authors:** Lisa Zirjacks, Nicolai Stransky, Lukas Klumpp, Lukas Prause, Franziska Eckert, Daniel Zips, Sabine Schleicher, Rupert Handgretinger, Stephan M. Huber, Katrin Ganser

**Affiliations:** 1Department of Radiation Oncology, Eberhard-Karls University, 72076 Tübingen, Germany; lisa.zirjacks@student.uni-tuebingen.de (L.Z.); Nicolai.Stransky@med.uni-tuebingen.de (N.S.); lukas.klumpp@gmx.net (L.K.); lukas.prause@student.uni-tuebingen.de (L.P.); Franziska.eckert@med.uni-tuebingen.de (F.E.); daniel.zips@med.uni-tuebingen.de (D.Z.); Katrin.ganser@med.uni-tuebingen.de (K.G.); 2Department of Hematology and Oncology, University Hospital Tuebingen, Children’s Hospital, 72076 Tuebingen, Germany; sb.schleicher@outlook.de (S.S.); rupert.handgretinger@med.uni-tuebingen.de (R.H.)

**Keywords:** brain tumor, primary spheroid culture, tumor-initiating cells, propidium iodide, Nicoletti staining, survival fraction, radioresistance, MGMT promoter methylation

## Abstract

Mesenchymal glioblastoma stem cells (GSCs), a subpopulation in glioblastoma that are responsible for therapy resistance and tumor spreading in the brain, reportedly upregulate aldehyde dehydrogenase isoform-1A3 (ALDH1A3) which can be inhibited by disulfiram (DSF), an FDA-approved drug formerly prescribed in alcohol use disorder. Reportedly, DSF in combination with Cu^2+^ ions exerts multiple tumoricidal, chemo- and radio-therapy-sensitizing effects in several tumor entities. The present study aimed to quantify these DSF effects in glioblastoma stem cells in vitro, regarding dependence on ALDH1A3 expression. To this end, two patient-derived GSC cultures with differing ALDH1A3 expression were pretreated (in the presence of CuSO_4_, 100 nM) with DSF (0 or 100 nM) and the DNA-alkylating agent temozolomide (0 or 30 µM) and then cells were irradiated with a single dose of 0–8 Gy. As read-outs, cell cycle distribution and clonogenic survival were determined by flow cytometry and limited dilution assay, respectively. As a result, DSF modulated cell cycle distribution in both GSC cultures and dramatically decreased clonogenic survival independently of ALDH1A3 expression. This effect was additive to the impairment of clonogenic survival by radiation, but not associated with radiosensitization. Of note, cotreatment with temozolomide blunted the DSF inhibition of clonogenic survival. In conclusion, DSF targets GSCs independent of ALDH1A3 expression, suggesting a therapeutic efficacy also in glioblastomas with low mesenchymal GSC populations. As temozolomide somehow antagonized the DSF effects, strategies for future combination of DSF with the adjuvant standard therapy (fractionated radiotherapy and concomitant temozolomide chemotherapy followed by temozolomide maintenance therapy) are not supported by the present study.

## 1. Introduction

Disulfiram (1-(diethylthiocarbamoyldisulfanyl)-n,n-diethyl-methanethioamide), also known under its trade name “Antabuse”, is an FDA-approved drug formerly prescribed in alcohol use disorder. By inhibiting aldehyde dehydrogenases (ALDH) of the liver, disulfiram leads to the accumulation of acetaldehyde after ethanol intake, resulting in severe hangover symptoms. Beyond sensitizing to alcohol, preclinical in vitro and animal studies demonstrated a tumoricidal, chemo- and/or radio-therapy-sensitizing (for review see [1]) as well as antitumor immune-response boosting activity [2,3] of disulfiram in several tumor entities. Among those are melanoma [4], non-small-cell lung cancer (NSCLC) [5], liver cancer [6], prostate cancer [7], pancreatic cancer [8], breast cancer [9], head and neck squamous cell carcinoma (HNSCC) [10], atypical teratoid/rhabdoid tumors [11], and glioblastoma [12,13]. Due to the preclinical evidence for an antitumor effect of disulfiram, several clinical trials with glioblastoma patients (ClinicalTrials.gov identifiers NCT03363659, NCT01777919, NCT01907165, NCT02715609, NCT03151772, NCT03034135, NCT02678975, NCT02770378) have been initiated, are ongoing or finalized (e.g., [14]).

Glioblastoma is, among primary brain tumors in adults, the most common and most malignant entity with very poor prognosis. Standard trimodal therapy comprises surgical resection, fractionated radiotherapy and concomitant temozolomide chemotherapy, followed by temozolomide maintenance therapy [15]. In addition to radio- and temozolomide resistance, the infiltrative, invasive growth of the tumor promotes therapy failure. The dissemination of glioblastoma cells in the brain parenchyma decreases the probability of complete tumor resection or coverage of all residual glioblastoma cells by the target volume of fractionated radiotherapy.

Glioblastoma omics data suggest distinct (e.g., classical, proneural and mesenchymal [16]) molecular subclasses. Among these, tumors with upregulated mesenchymal expression or methylation patterns associate with the worst prognosis [17,18,19,20,21]. The mesenchymal profile results in part from the prevalence of mesenchymal glioblastoma stem (cell-like) or tumor-initiating cells in these tumors [22]. This cell subpopulation has been associated with tumor spreading.

Reportedly, transition of carcinoma cells into hybrid epithelial–mesenchymal cells is likely associated with the acquisition of stemness and precedes tumor metastasis [23]. Likewise, mesenchymal glioblastoma stem cells, which constitute a minor subpopulation of glioblastoma cells, are held responsible for glioblastoma spreading in the brain and formation of distant secondary lesions [22,24]. Thus, eradication of mesenchymal glioblastoma stem cells might be a prerequisite to control glioblastomas of the mesenchymal subclass. ALDH1A3 reportedly plays a pivotal role in the maintenance of stemness in mesenchymal cancer stem cells [8,25]. Via acting on ALDH1A3 disulfiram might specifically target mesenchymal glioblastoma stem cells. Pharmacokinetics data, however, indicate rapid metabolization of disulfiram. Moreover, therapeutically achievable concentrations of disulfiram in the brain might be low, and tumoricidal actions of disulfiram seem to be mediated rather by its Cu^2+^-overloading than its ALDH-inhibiting function as introduced in the next paragraphs. 

In the acid environment of the stomach, ingested disulfiram is reduced to two molecules of diethyldithiocarbamate that form hydrophobic bis-(diethyldithiocarbamate)-Cu(II) complexes. The latter and uncleaved disulfiram are readily absorbed by the gastrointestinal tract. In the blood, the erythrocytic glutathione reductase may split the bis-(diethyldithiocarbamate)-Cu(II) complexes into diethyldithiocarbamate monomers which form mixed disulfides with free thiols of proteins (for review see [26]). Moreover, disulfiram entering the blood may be alternatively reduced by a reaction with serum albumin to diethyldithiocarbamate and mixed disulfide of diethyldithiocarbamate with serum albumin [27]. Beyond binding to plasma proteins, diethyldithiocarbamate entering the liver may become S-methylated to methyl-diethyldithiocarbamate by thiopurine or thiol methyltransferase [28], and S-oxidized by microsomal cytochrome P450 monooxygenase to the corresponding sulfoxide and sulfone. The latter have been proposed to play an important role in forming inhibitory covalent cysteine adducts with aldehyde dehydrogenases (ALDHs) (for review see [26]).

The maximal dose of disulfiram tolerated by glioblastoma patients in combination with chemotherapy was 500 mg *p.o.*, once daily [29]. Pharmacokinetic data suggest that a single oral dose of 500 mg gives rise to mean peak total plasma concentrations of disulfiram (t_1/2_ = 7.3 h [30]) and its metabolites diethyldithiocarbamate and methyl-diethyldithiocarbamate between 0.5 and 2 µM around 6–10 h after ingestion with very high interpatient variability [31]. As disulfiram and metabolites are either lipophilic or highly reactive, the overwhelming majority of these molecules can be speculated to bind to serum albumin, profoundly lowering their free plasma concentrations. Diethyldithiocarbamate is detoxified by rapid glucuronidation and renal excretion, or is decomposed into diethylamine and carbon disulfide that are excreted or exhaled (for review see [26]).

Disulfiram (and probably most metabolites) permeates the blood–brain barrier [32], suggesting that the interstitial concentrations of disulfiram and metabolites in the brain is in equilibrium with the unbound (un-glucuronidated) free plasma pool of these compounds. If so, and if there are not any specific processes leading to their accumulation, interstitial brain concentrations of disulfiram and metabolites can be expected to be far below 1 µM. This should be considered when designing in vitro studies on the tumoricidal disulfiram effects in, e.g., glioblastoma. 

Several studies show that Cu^2+^ ions contribute to the tumoricidal effect of disulfiram (e.g., [7,12,33,34]). Mouse ^64^Cu PET- [35] and rat optical emission spectrometry studies [36] have demonstrated that disulfiram and diethyldithiocarbamate, respectively, increase Cu^2+^ transport into the brain most probably via formation of lipophilic bis(diethyldithiocarbamate)-Cu(II) complexes [36]. In the brain, cellular Cu^2+^ uptake occurs by lipid diffusion of these complexes across the plasma membrane. Alternatively, in an acidified brain-tumor microenvironment, uncharged, protonated diethyldithiocarbamate and Cu^+^ may enter cells separately via lipid diffusion and activated copper transporter 1, Ctr1, respectively [37].

Total Cu^2+^ ion concentrations up to 25 µM [38,39]) have been reported in blood serum of healthy persons. In blood, Cu^2+^ binds to ceruloplasmin, serum albumin, as well as enzymes and clotting factors (5%). Only a low fraction (0.2–2.5%) of Cu^2+^ forms small-molecular-weight (SMW) ternary complexes with histidine or other amino acids [39] suggesting blood SMW Cu^2+^concentrations in the range of 50–500 nM. In cerebrospinal fluid (CSF) with much lower Cu^2+^ protein buffer capacity, a total Cu^2+^ concentration of 160 nM has been described [40] which might hint to free interstitial brain Cu^2+^ concentrations of ≤100 nM. 

Disulfiram-provoked cellular Cu^2+^ overload induces the redox cycling of hydrogen peroxide to hydroxyl radicals (OH•) via the Harber–Weiss reaction. OH•, in turn, may peroxidize lipids or damage proteins and DNA [41]. This disulfiram/Cu^2+^-mediated impairment of redox homeostasis [33] is most probably the reason for the observed pleiotropic actions of disulfiram. Besides blockage of ALDH isoforms, disulfiram/Cu^2+^ reportedly modulate among others the proteasome [42], DNA-methyltransferases [43] including the O6-methylguanin-DNA-methyltransferase [44], the cystathionine-β-synthase [45], matrix metalloproteinases-2 and -8 [46], caspases [47], the EGFR/c-Src/VEGF-pathway [48], the NF-κB and TGF-β pathway [6], cell-matrix adhesion [49], lysosomal membrane integrity [50], immunogenic cell death [3], immunosuppression [2], as well as sensitivity to chemo- (e.g., [51]) and radio-therapy (e.g., [10]).

The complex degradation of disulfiram in pharmacologically active metabolites and their interplay with Cu^2+^ ions suggest that in vivo effects of disulfiram cannot easily be mimicked in cell culture systems. Indeed, the Cu^2+^ concentrations vary considerably between different cell culture media and may be unphysiologically low in fetal bovine serum-free media frequently used for induction or selection of stem cells, as used in the present study. Beyond exerting toxic redox effects, Cu^2+^ ions have been demonstrated to facilitate the reduction of disulfiram to diethyldithiocarbamate and formation of bis(diethyldithiocarbamate)-Cu(II) complexes in cell culture medium. This reaction seems to be slow (82% yield after 1 day) and might be a prerequisite for the reported in vitro inhibition of ALDH isoforms by disulfiram. This blockade probably involves an intramolecular disulfide bond between adjacent cysteines in the active site of the enzyme isoforms, resulting from unstable mixed disulfide interchange reactions [52]. Together, these observations suggest that the dual inhibitory action (Cu^2+^-mediated oxidative stress and ALDH inhibition) of disulfiram can be investigated in appropriately Cu^2+^-supplemented in vitro cell models.

The present study aimed to quantify in vitro the tumoricidal, temozolomide-, and radio-sensitizing function of disulfiram/Cu^2+^ on cell cycle distribution and clonogenic survival of isocitrate dehydrogenase (IDH) wildtype, O6-methylguanine-DNA-methyltransferase (MGMT) promoter-unmethylated, temozolomide-resistant glioblastoma stem cells grown in primary culture. In particular, the dependence of the disulfiram/Cu^2+^ effects on the mesenchymal stem-cell marker ALDH1A3 was addressed.

## 2. Material and Methods

### 2.1. Cell Culture

Primary LK7 and LK17 glioblastoma stem cells (pGSC) were grown from the resection specimens from two male glioblastoma patients (66 and 54 years of age) who underwent surgery at the Department of Neurosurgery at the University of Tübingen in 2016. Both tumors had unmethylated MGMT promoters and IDH wildtype alleles, suggesting comparable pathological characteristics. All experiments were approved by the local ethics commission (project #184/2015BO1 and #184/2015BO2). Cells were dissociated enzymatically and mechanically and plated in complete human NeuroCult NS-A Proliferation Medium including 20 ng/mL rhEGF, 10 ng/mL rhFGF and 2 ppm heparin (STEMCELL Technologies, Vancouver, BA, Canada) as described [53]. According to the information from the manufacturer, this medium contains 1 nM Cu^2+^ ions (personal communication). Albeit grown in identical medium and cell culture flasks, LK7 cells formed adherent monolayers while LK17 grew in free-floating spheroids. LK17 cells were isolated from spheroids and adherent LK7 cells detached by accutase digestion (5 min, 37 °C; Sigma Aldrich, Taufkirchen, Germany). To “differentiate” pGSCs into “bulk” tumor cells, LK7 and LK17 cells were grown for 5 days in RPMI-1640 medium containing 10% fetal bovine serum (FBS) (Sigma Aldrich). The applied inhibitors (CuSO_4_, temozolomide, disulfiram, and diethylaminobenzaldehyde) as well as the solvent dimethyl sulfoxide (DMSO) were from Sigma Aldrich.

### 2.2. RNA Isolation and Real-Time Reverse-Transcriptase Polymerase Chain Reaction (RT-PCR)

RNA was isolated and fragments specific for *ALDH1A3* (QT00077588, QuantiTect Primer Assay, QIAGEN, Venlo, The Netherlands), *SOX2* (QT00237601), *NOTCH1* (QT01005109), Musashi-1, (*MSI1*, QT00025389), fatty acid-binding protein-7 (*FABP7*, QT00007322), prominin-1, (*PROM1*, CD133, QT00075586), as well as for the housekeepers glyceraldehyde-3-phosphate dehydrogenase (*GAPDH*, QT01192646), β-actin (*ACTB*, QT00095431) and pyruvate dehydrogenase-E1 subunit β (*PDHB*, QT00031227) were amplified by SYBR Green-based real-time PCR (1Step RT qPCR Green ROX L Kit, highQu, Kraichtal, Germany), as described [53].

### 2.3. Immunoblotting

Total protein was lysed in a buffer containing 25 mM Tris/NaOH (pH 7.6), 150 mM NaCl, 35 mM sodium dodecyl sulphate (SDS), 12 mM sodium deoxycholate, additionally supplemented with Triton x-100 (1.0%), proteinase inhibitor (complete tablets Mini EDTA-free, Roche, Basel, Switzerland), and phosphatase inhibitor cocktail-2 (Sigma Aldrich), diluted 3:4 in 4× sample buffer containing 14 mM SDS, 200 mM Tris/NaOH (pH 6.8), 10 mM dithiothreitol further containing glycerol (20%) and bromophenol blue (0.005%), and heated (5 min, 70 °C). Proteins were separated by SDS-polyacrylamide gel (10%) electrophoresis, blotted, and probed against ALDH1A3 and GAPDH using primary rabbit polyclonal l anti-ALDH1A3 antibody (#PA5-29188, batch number TH2617862, 1:1000, Thermo Scientific, Waltham, MA, USA), and mouse monoclonal anti-GAPDH antibody (clone 6C5, #ab8245, 1:20,000, Abcam, Cambridge, MA, USA), respectively. Antibody binding was detected after binding of horseradish peroxidase (HRP)-linked antirabbit IgG (#70749, 1:2000, Cell Signaling Technology, Frankfurt a. Main, Germany) and antimouse IgG (#NA931V, 1:2000, GE Healthcare, Buckinghamshire, UK) secondary antibody, respectively, with Immobilon Western HRP substrate (Merck Millipore, Darmstadt, Germany) and recorded with ChemiDoc Imaging System (BioRad Laboratories, Feldkirchen, Germany). 

### 2.4. ALDH Activity

The ALDEFLUOR Kit (STEMCELL Technologies) was used to determine ALDH activity. Exponentially growing LK7 monolayers and LK17 spheroides (8–32 cell stage), were detached/isolated and incubated (3 × 10^5^ cells/500 µL assay buffer for 30 min at 37 °C) in complete NeuroCult medium containing the fluorescent substrate bodipy-aminoacetaldehyde and 100 nM CuSO_4,_ further containing dimethylsulfoxide (DMSO, 0.1%, vehicle control) and the ALDH inhibitor diethylaminobenzaldehyde (DEAB, 0 or 3 µM) or disulfiram (0 or 100 nM). ALDH-dependent conversion of the substrate into intracellularly trapped bodipy-aminoacetate was measured by flow cytometry (FACScalibur with CellQuest software, BD, Franklin Lakes, NJ, USA) at 488 nm excitation and 530/30 nm emission wavelength and analyzed by FCS Express-3 software (version 3.00.0825, De Novo Software, Pasadena, CA, USA).

### 2.5. Cell Cycle Analysis in Flow Cytometry

Detached/isolated LK7 and LK17 pGSC cells were grown for 3–4 days, preincubated (30 min), irradiated (0, 4 or 8 Gy) by 6 MV photons with a linear accelerator (LINAC SL15, Philips, Einthoven, The Netherlands) at a dose rate of 4 Gy/min at room temperature, and incubated for further 48 h at 37 °C in complete NeuroCult medium supplemented with 100 nM CuSO_4_, further containing DMSO (0.1% vehicle control) and disulfiram (0 or 100 nM) or temozolomide or both (0 or 30 µM). 

For cell cycle analysis, cells were detached/isolated, permeabilized and stained (30 min at room temperature) with Nicoletti propidium iodide solution (containing 0.1% Na-citrate, 0.1% triton X-100, 10 µg/mL propidium iodide in phosphate-buffered saline, PBS), and the DNA quantity was analyzed by flow cytometry (FACScalibur, BD, Franklin Lakes, NJ, USA) at 488 nm excitation and 585/40 nm emission and analyzed by FCS Express-3 software.

### 2.6. Clonogenic Survival of Irradiated Cells

Single-cell suspensions of LK7 and LK17 cells were sequentially 1:2 diluted in 96-well plates resulting in 12 cell dilutions (2048 to 1 cell(s)) per well in 100 µL complete NeuroCult medium (or 10% FBS-containing RPMI medium for Figure 1D only) and sedimented overnight. Then, cells were preincubated (1 h), irradiated (0, 4 or 8 Gy), and postincubated (4 weeks) in complete NeuroCult medium supplemented with 100 nM CuSO_4_, further containing DMSO (0.1% vehicle control) and disulfiram (0 or 100 nM, and for initial dose-finding experiments also with 1000 nM and 10,000 nM) or temozolomide or both (0 or 30 µM). 

Thereafter, minimal cell number required to restore the culture (LK7) or required for spheroid formation (LK17) was determined. The reciprocal value of this minimal number defined the plating efficiency (PE). To calculate the survival fractions (SF), the PEs at the different radiation doses were either normalized to the mean PE of the 0 Gy/vehicle control (Figures 4B and 5B) or of the corresponding 0 Gy controls (Figures 4C,D and 5C,D) according to the equation: SF_0–8 Gy_ = PE_0–8 Gy_/PE_0 Gy_. The survival fractions (SF) thus obtained were plotted against the radiation dose (d) and fitted according to the linear quadratic model with the following equation derived from the linear quadratic model: SF = e^−(α·d + β·d^2^), with α and β being cell-type-specific parameters.

### 2.7. Statistics

Data are shown as individual values or means ± SE. Differences between two sample groups were assessed by Welch-corrected unpaired two-tailed *t*-test (Figure 1D and Figures 2B and 3B,C). Differences between more than two sample groups (Figures 3D and 4–7) were evaluated by nonparametric Kruskal–Wallis with Dunn’s multiple comparison test. Error probabilities of *p* ≤ 0.05 were assumed to indicate statistical significance. Statistical tests were performed with GraphPad Prism (version 8.4.0, GraphPad Software, La Jolla California, CA, USA).

## 3. Results

Despite identical conditions, primary cultures of glioma stem cells (pGSCs) show different growth phenotypes ranging from free-floating spheroids to adherent monolayers [53]. In particular, LK7 pGSCs grew in complete NeuroCult stem cell (NSC) medium as an attached monolayer while LK17 pGSCs formed adherent spheroids (Figure 1A) with doubling times of about 1.0 (LK17) and 1.7 (LK7) days (Figure 1B,C). On the mRNA level, LK7 and LK17 cells differed in their abundances of stem-cell markers. While the mRNA encoding the mesenchymal stem-cell marker ALDH1A3 was much more abundant in LK7 than in LK17, mRNAs of the stem-cell markers Musashi-1 (*MSI1*) and Prominin-1 (*PROM1*, CD133) were markedly higher in LK17 than in LK7 pGSCs. The proneural (*NOTCH1*, *SOX2*) and glial (*FABP7*) stem-cell marker mRNAs, in contrast, were similarly abundant in both pGSCs (Figure 1D, open columns). 

“Differentiating” the pGSC into “bulk” glioblastoma cells by changing the medium to 10% FBS-containing RPMI 1640 resulted in a dramatic decrease of plating efficiencies in both pGSCs (Figure 1D). In addition, FBS “differentiation” was paralleled in LK7 by a downregulation of *ALDH1A3*, *SOX2*, *MSI1* and *FAPB7* mRNA and in LK17 cells by a decrease in *NOTCH1*, *SOX2*, *MSI1*, *PROM1* and *FABP7* (the latter two did not reach statistical significance) as well in an increase of *ALDH1A3* mRNA abundance (Figure 1E, compare open and closed columns). Moreover, FBS “differentiation” induced in LK17 cells a change in growth morphology from spheroid to adherent monolayer growth (data not shown). Together, the increase in plating efficiency as a measure of self-renewal capability and clonogenicity and the enrichment of stem-cell markers by cultivation in FBS-free NeuroCult (NSC) medium points to an enrichment of GSCs by induction or selection of GSCs in NSC-containing medium when compared to FBS-containing medium. This was also suggested by the fact that LK7 (LK17 were not tested) developed orthotopic glioblastoma when transplanted into the right striatum of immunocompromised mice (data not shown) indicating their tumor-initiating capability. Finally, the differing profiles of stem-cell marker abundances suggest that LK7 and LK17 harbor different GSC subpopulations. 

Next, we tested, in the continuous presence of CuSO_4_ (100 nM), the sensitivity of our pGSCs in NSC medium to various concentrations (100 nM–10 µM) of disulfiram by using clonogenic survival as the endpoint (Figure 2A). In both pGSCs, the IC_50_ for disulfiram was below 100 nM. Since disulfiram in the range of 100 nM is expected to be achieved in the brain upon oral prescription (see Introduction section) and since this concentration already evoked a pronounced reduction of clonogenicity in our pGSCs (Figure 2A), we applied 100 nM disulfiram (together with 100 nM CuSO_4_) in all further experiments.

To study the effect of disulfiram/Cu^2+^ (24 h) on the stemness properties of our pGSCs, the changes in mRNA abundance of the stem-cell markers *ALDH1A3*, *NOTCH1*, *SOX2*, *MSI1*, *PROM1*, and *FABP7* were analyzed. Beyond decline in clonogenic survival, disulfiram/Cu^2+^ either did not alter or induced (*NOTCH1*, *MSI1*) expression of stem-cell-marker-encoding mRNAs in LK7 cells. (Figure 2B). In LK17 cells, in sharp contrast, disulfiram/Cu^2+^ treatment showed a trend (*p* values between 0.12–0.21, two-tailed Welch-corrected *t*-test) to reduce abundances of all tested marker mRNAs except that of *ALDH1A3* (the latter increased significantly at a very low level, Figure 2B). Combined, these data suggest that disulfiram-mediated inhibition of clonogenicity may be associated with up or downregulation of stemness markers. In particular in LK7 cells, disulfiram treatment seemed to induce rather than downregulate stemness.

According to our previous findings (see Figure 1D and Figure 2B), LK7 and LK17 differed in their *ALDH1A3* mRNA abundance. To directly compare mRNA abundance with protein and functional expression of this mesenchymal stem-cell marker in NSC medium between both pGSCs, we conducted a further set of experiments applying RT-PCR, whole lysate immunoblotting and flow cytometry (Figure 3). The profoundly higher *ALDH1A3* mRNA abundance (Figure 3A) was paralleled by a >10-fold higher ALDH1A3 protein abundance in LK7 when compared with LK17 pGSCs (Figure 3B,C). Consistently with this difference, DEAB-sensitive enzymatic activities of the ALDH isoforms were higher in LK7 compared with LK17 cells when measured in the presence of CuSO_4_ (100 nM) under all experimental conditions by flow cytometry (Figure 3D,E, black and blue). Notably, disulfiram exerted only an incomplete blockage of ALDH activity (Figure 3D,E, red). Together, these data point to a mesenchymal phenotype of the LK7 pGSC but not of LK17 cells.

To test for effects of disulfiram alone or in combination with radiation and/or temozolomide chemotherapy on cell cycle and cell death, we pretreated (1 h) exponentially growing LK7 monolayers and LK17 spheroides (8–32 cell stage) in NSC medium, irradiated them with 0–8 Gy, and postincubated the cells for 48 h with vehicle alone (0.1% DMSO), with disulfiram (100 nM), with temozolomide (30 µM), or with disulfiram and temozolomide (additionally, the medium of all experimental arms contained CuSO_4_, 100 nM). Thereafter, cellular DNA content was analyzed by propidium iodide binding in flow cytometry (Figure 4A).

In 8 Gy- but not 0 or 4 Gy-irradiated LK7 cells, disulfiram and temozolomide as well their combination increased sub-G_1_ fraction (i.e., cells with partially degraded DNA, Figure 4B, left) in an apparently nonadditive manner in about 1% of cells. This points to a disulfiram- and temozolomide-induced death of cells injured by radiation in a small cell fraction. An increase in cellular DNA content caused by errors in mitosis (polyploidy) or cytokinesis (polynuclearity), in contrast, was not observed in LK7 in either treatment arm, as evident from unchanged hyper-G fraction (Figure 4B, right).

Irradiation with 8 Gy evoked a G_2_/M cell cycle arrest in LK7 cells as evident from a decrease in G_1_ and S phase population and an increase in G_2_ population (closed circles in Figure 4C). Independent of irradiation, disulfiram showed a tendency to decrease G_1_ and increase G_2_ population (Figure 4C, left and right). Moreover, disulfiram induced almost a doubling of S population especially in irradiated cells (Figure 4C, middle). Notably, temozolomide, which did not exert any effect on cell cycle as monotreatment, seemed to mitigate the disulfiram effects in combined application (Figure 4C).

Similar to LK7, disulfiram decreased G_1_ and increased G_2_ population in LK17 cells independent of irradiation (Figure 5A,B, left and right). In contrast to LK7, disulfiram treatment did not change S population here (Figure 5B, middle). Likewise, temozolomide as a monotreatment induced an increase in G_1_ (8 Gy) and decrease in G_2_ (4 Gy and 8 Gy) population but only in irradiated cells (Figure 5B, left and right, open triangles). Again, the temozolomide and disulfiram effects were not additive. Instead, temozolomide seemed to attenuate the disulfiram effect in combined application as evident from the 0 Gy and 4 Gy data in Figure 5B, right (open diamonds). In control or irradiated LK17 cells, disulfiram or temozolomide did not increase sub-G_1_ or hyper-G populations (data not shown). Combined, these data suggest some interference with the cell cycle by disulfiram in LK7 and LK17 and by temozolomide in LK17 cells. These effects, however, did not translate to pronounced cell death (sub-G_1_ population) or impairment of mitosis/cytokinesis (hyper-G population) during the 48 h period of observation.

To test for effects on clonogenic survival, LK7 and LK17 cells were detached/isolated, sequentially 1:2 diluted (2048 to 1 cell(s) per well) in NSC medium in 96-well plates, sedimented overnight, preincubated (1 h), irradiated (0–8 Gy), and postincubated (4 weeks) with vehicle alone (0.1% DMSO), with disulfiram (100 nM), with temozolomide (30 µM), or with disulfiram and temozolomide. Again, CuSO_4_ (100 nM) was added to the medium in all experimental arms. 

Plating efficacy was defined by the reciprocal of the minimal cell number required to regrow culture (LK7) or to form spheroids (LK17). Survival fractions were calculated by normalizing plating efficiencies either to that of the 0 Gy vehicle control or to the respective 0 Gy control of each experimental arm. The former data representation illustrates potential additive effects of radiation and disulfiram or temozolomide, and the latter reveals potential radiosensitizing or radioresistance-conferring effects of the drugs.

As shown in Figure 6A, disulfiram profoundly lowered the plating efficiency of LK7 cells. Temozolomide, in contrast, did not significantly decrease LK7 plating efficiency but attenuated the effect of disulfiram when both drugs were coapplied. This can also be deduced from Figure 6B, where the plating efficiencies were normalized to the 0 Gy vehicle control and the survival fractions were plotted against the radiation dose: temozolomide, which alone only marginally alleviated survival fractions, markedly attenuated the inhibitory disulfiram effect at all tested radiation doses when both drugs were simultaneously administrated. Normalizing the plating efficiencies to the 0 Gy control of the respective experimentation arm (Figure 6C) indicated that neither disulfiram nor temozolomide nor their combination exerted any radioresistance-modifying function in LK7 cells. This is also demonstrated in more detail by the individual survival fractions after irradiation with 4 Gy (SF_4Gy_), as depicted in Figure 6D. 

Finally, plating efficiencies and survival fractions were determined in 0–8 Gy-irradiated LK17 cells (Figure 7). Identically to LK7, disulfiram strongly attenuated plating efficiencies in LK17 cells which were insignificantly blunted by co-administration of temozolomide, while temozolomide alone had no statistically significant effect (Figure 7A,B). Additionally, in analogy to LK7, neither disulfiram nor temozolomide nor their combination radiosensitized LK17 cells (Figure 7C,D). 

Taken together, these datasets indicate high inhibition of clonogenic survival by disulfiram in glioblastoma stem cells, independent of ALDH1A3 expression. In addition, temozolomide exerted no statistically significant inhibitory effects on clonogenic survival, but strongly mitigated the disulfiram effect in LK7 cells. Finally, disulfiram and temozolomide failed to radiosensitize LK7 or LK17 cells, neither as monotreatment nor in combination.

## 4. Discussion

Repurposing the FDA-approved ALDH blocker disulfiram for anti-glioblastoma treatment has been proposed as a promising strategy to overcome therapy resistance. Preclinical evidence that glioblastoma patients might benefit from an implementation of disulfiram concomitant to the standard therapy protocol—that is, in the case of glioblastoma adjuvant temozolomide radiochemotherapy and maintenance therapy—is limited. Therefore, the scope of the present study was to analyze in a clinically relevant cell model, i.e., in temozolomide-resistant primary glioblastoma stem-cell cultures, the potential temozolomide- and radio-sensitizing function of disulfiram. Moreover, by comparing two glioblastoma stem-cell subpopulations that differ in ALDH activity, this study addressed the question of whether disulfiram may specifically target ALDH-expressing mesenchymal glioblastoma stem cells.

### 4.1. Disulfiram as Anti-Glioblastoma Agent and Temozolomide Sensitizer

Several in vitro studies have demonstrated a tumoricidal effect of disulfiram in various tumor entities including glioblastoma [12,54]. In particular, temozolomide-refractory glioblastoma (stem) cells have been demonstrated to be sensitive to disulfiram [54]. Moreover, a chemotherapy-sensitizing action of disulfiram has been reported: disulfiram/Cu^2+^ sensitizes temozolomide-resistant glioblastoma cells to temozolomide in vitro [12,54] and in an orthotopic glioblastoma mouse model (daily 100 mg/kg B.W. disulfiram and 2 mg/kg B.W. Cu^2+^) [12].

Temozolomide is a DNA-alkylating agent that methylates purine bases of the DNA at position O6 and N7 of guanine and N3 of adenine [55]. O6-methylguanine (O6-meG) is assumed to be the most hazardous DNA modification that may lead to O6-meG/T mispair-mediated mutagenesis, or more importantly, to cytotoxic DNA double-strand breaks (DNA DSBs). The latter result from futile repair cycles of the mismatch repair (MMR) system during two rounds of DNA replication [56,57]. MMR deficiency as well as O6-methylguanine-DNA methyltransferase (MGMT) confer resistance against temozolomide. MGMT directly demethylates O6-meG and is downregulated in about 45% of glioblastoma patients with MGMT promoter methylation in the tumor and enhanced temozolomide sensitivity [15]. 

A reported mechanism of temozolomide chemosensitization by disulfiram has been identified in pituitary adenoma stem-like cells [51] and in glioblastoma cell lines [44]: disulfiram covalently modifies MGMT, leading to the proteasomal degradation of the DNA repair enzyme. In addition, disulfiram has been proposed in glioblastoma spheroid cultures to facilitate the DNA-damaging temozolomide effect by impairing DNA repair [12]. 

Temozolomide-mediated DNA DSBs reportedly trigger a G_2_/M arrest of cell cycle [55]. In our present experiments (see Figure 4 and Figure 5), a temozolomide-mediated G_2_/M arrest could not be detected in unirradiated LK7 and LK17 cells. Given the doubling times of exponentially growing LK7 and LK17 pGSCs in NSC medium of 1.7 and 1.0 days, respectively, (see Figure 1C) it can be assumed that all cells (LK17) or a significant fraction of cells (LK7) underwent two rounds of DNA replication (required for temozolomide-triggered MMR-mediated DNA damage) during the chosen incubation period (48 h) of the flow cytometry experiments (see Figure 4 and Figure 5). Moreover, temozolomide at the chosen concentration (30 µM) has been demonstrated in our previous experiments to exert a high tumoricidal effect in MGMT promotor-methylated pGSCs (unpublished own observations). Thus, the flow cytometry data on cell cycle and cell death of the present study confirms the relative temozolomide resistance of MGMT promoter-unmethylated glioblastoma. This was also evident from the statistically insignificant effects of temozolomide on clonogenic survival in both pGSC cultures (see Figure 6A and Figure 7A).

While confirming the tumoricidal action of disulfiram/Cu^2+^ in temozolomide-resistant glioblastoma stem-cell cultures, our present study did not observe a temozolomide-sensitizing effect of disulfiram/Cu^2+^ (see Figure 6A and Figure 7A). Quite the contrary, in both cell models, temozolomide markedly or had a tendency to attenuate the inhibitory effect of disulfiram on clonogenic survival. Such a disulfiram effect-diminishing action of temozolomide was also suggested by our flow cytometry experiments on the cell cycle (see Figure 4 and Figure 5). One might speculate that temozolomide interferes with lethal pathways triggered by disulfiram. Independent of the underlying molecular mechanisms, the present observations do not support future therapy strategies pursuing a concomitant disulfiram–temozolomide chemotherapy. In addition, this observation suggests that the tumoricidal effect of disulfiram may be sensitive to pharmaco-interactions with co-medications. The understanding of such pharmaco-interactions, however, is a prerequisite for the success of future clinical trials using disulfiram for second-line therapy in glioblastoma patients with tumor progression during temozolomide maintenance therapy. The analysis of the molecular mechanism of such pharmaco-interactions (here, the temozolomide-disulfiram interaction), however, goes beyond the scope of the present study.

### 4.2. Disulfiram as a Radiosensitizer 

Likewise, our present study did not identify any radiosensitization of both glioblastoma stem-cell cultures by disulfiram/Cu^2+^. This is in seeming contrast to previous studies that show a disulfiram/Cu^2+^-mediated radiosensitization in patient-derived spheroid glioblastoma stem (brain-tumor-initiating) cells [12] and human glioblastoma cell lines [58]. Notably, in the latter study, only one (U138MG) and in tendency also a second (T98G) out of five glioblastoma lines were radiosensitized by disulfiram (75–100 nM) when grown in Cu^2+^-containing serum-supplemented medium and when using clonogenic survival as the endpoint [58]. Clonogenic survival determines the probability of a treated tumor to relapse, and is therefore thought to be the gold standard for the interpretation of drug effects on radiosensitivity in radiation biology [59]. In the glioblastoma stem-cell spheroid cultures, 5 Gy irradiation in combination with disulfiram (100 nM) and Cu^2+^ (200 nM) further decreased viability (as defined by metabolic activity and compared to the disulfiram/Cu^2+^/0 Gy arm) of only one out of two tested spheroid cultures [12]. In addition, in the same study, disulfiram/Cu^2+^ delayed repair of DNA double-strand breaks (DSBs) of 2 Gy-irradiated cells without increasing the number of residual (24 h-value) DSBs, as analyzed by the counting of nuclear γH2AX (phosphorylated histone H2AX) foci [12]. 

Since only limited conclusions on clonogenic survival can be drawn from the decay of radiation-induced γH2AX foci [60] as well as metabolically defined “viability” of irradiated cancer cells, the reported evidence for a radiosensitizing function of disulfiram in glioblastoma stem cells is limited. Combined with the notion that disulfiram radiosensitized only a minor fraction of the tested panel of glioblastoma cell lines [58], and additionally considering the results of our present study, it can be concluded that disulfiram may radiosensitize glioblastoma (stem) cells, but this seems to be rather an exception than a general phenomenon. The situation is different in irradiated AT/RT (atypical teratoid/rhabdoid) brain tumor lines and primary cultures, where disulfiram (in Cu(II)-containing serum-supplemented medium) consistently decreases survival fractions in colony formation assays of all tested cell models with an EC_50_ of 2–50 nM [61].

### 4.3. Cu^2+^-Mediated Oxidative Stress

The radiosensitizing action of disulfiram probably depends on the Cu^2+^ ion-overloading function of the drug. Ionizing radiation induces beyond instant radical formation (e.g., formation of OH• by ionization of H_2_O) delayed long-lasting mitochondrial-generated superoxide anion (O_2_^−^•) formation which contributes to radiation-mediated genotoxic damage [62]. It is tempting to speculate that disulfiram-mediated Cu^2+^ overload and subsequent OH• formation (see introduction) collaborates with radiation-triggered mitochondrial oxidative stress (and also with temozolomide) in introducing DNA DSBs. If so, the radiosensitizing (and also temozolomide-sensitizing) effect of disulfiram should be, on the one hand, a direct function of the interstitial Cu^2+^ concentration, and on the other, a function of the intracellular Cu^2+^-reducing, Cu^+^-chaperoning, -sequestrating, and -extruding capability as well as the oxidative defense of a tumor cell [63,64]. The Cu^2+^-detoxifying capability most probably differs between cell types, and might explain the difference in reported radiosensitizing activity of disulfiram between AT/RT [61] and the glioblastoma (stem) cells ([12,59] and present study). In particular, tumor stem cells have been demonstrated to exhibit upregulated drug-efflux pumps, DNA repair, and oxidative defense [65]. 

### 4.4. Does Disulfiram Specifically Eradicate Mesenchymal Glioblastoma Stem Cells? 

In an orthotopic mouse model of human glioblastoma, disulfiram inhibited formation of micrometastasis [13]. Moreover, a high-throughput screen in FBS-free NSC medium identified, via viability assay, disulfiram as a potent growth inhibitor (mean IC_50_s of 12–56 nM) of patient-derived glioblastoma stem cells [34]. Of note, chelation of Cu^2+^ decreased and addition of Cu^2+^ to the medium increased the disulfiram effect in this high-throughput screen. Similarly, the disulfiram-mediated inhibition of ALDH-positive glioblastoma stem cells has been demonstrated to depend on Cu^2+^ [66]. Along those lines, disulfiram diminished clonogenic survival of glioblastoma stem cells in an ALDH(1A3)-independent manner in our present study. Together, these findings suggest that disulfiram equally targets mesenchymal and nonmesenchymal glioblastoma stem cells, and that ALDH inhibition by disulfiram does not play a role herein.

The disulfiram concentration (100 nM) applied in our work was above the IC_50_ concentration for blockage of clonogenic survival in both pGSCs (see Figure 2A). Such a low IC_50_ is in good agreement with those reported for GSCs in NSC medium [34], as mentioned above. In FBS-containing medium, higher IC_50_ values (120–465 nM [66]) for disulfiram have been observed in glioblastoma cell lines. This might point to a lowering of the free disulfiram concentration by binding to FBS, aggravating the direct comparison of in vitro data obtained under different culture conditions. Nevertheless, submicromolar IC_50_ values indicate potent tumoricidal effects of disulfiram in vitro, which is in sharp contrast to the disappointing outcome of clinical trials.

### 4.5. Disulfiram in Clinical Trials

Recent clinical trials on newly diagnosed [29] and recurrent glioblastoma ([14,67]) tested disulfiram together with dietary Cu^2+^ supplementation during alkylating chemotherapy. The data analyses so far suggest feasibility of disulfiram/Cu^2+^ treatment during chemotherapy but do not indicate any temozolomide-sensitizing or tumoricidal action of disulfiram in glioblastoma [14,29]. Likewise, a clinical trial in men with nonmetastatic, recurrent prostate cancer after local therapy did not show a clinical benefit of disulfiram (250 or 500 mg daily) [68]. In addition, epidemiological data did not identify any associations between incidence of melanoma, breast, or prostate cancer and long-term disulfiram use [69].

This apparent discrepancy to the strong tumoricidal effect of disulfiram observed in preclinical studies might suggest that in the clinical setting, therapeutically effective disulfiram (Cu^2+^) concentrations are not reached in the tumors. Encapsulation of disulfiram in polymeric nanoformulations, micelles, microparticles, nanocrystals or lipid-based drug delivery systems might be approaches in the future to improve the pharmacokinetic profile of disulfiram in patients [70]. Moreover, surface receptor-specific targeting of disulfiram-bearing nanoparticles might enhance tumor specificity and cellular drug uptake of disulfiram therapy [71]. Alternatively, tumor specificity may be attained by specific application routes such as delivering disulfiram to the brain via nasally applied nanoemulsion [72] or stereotactic injection [73].

### 4.6. Concluding Remarks

The present study disclosed a strong tumoricidal effect of disulfiram/Cu^2+^ in primary cultures of ALDH1A3^+^ and ALDH1A3^−^ glioblastoma stem cells. In contrast to previous studies, neither a radiotherapy- nor a temozolomide-sensitizing action of disulfiram was observed. Temozolomide, on the contrary, attenuated the tumoricidal disulfiram effect. Translated into the clinical situation, our observations do not support the combination of disulfiram with adjuvant standard glioblastoma therapy (fractionated radio-temozolomide therapy and subsequent temozolomide maintenance therapy). Disulfiram/Cu^2+^ might be tested in future strategies for second-line chemotherapy in glioblastoma if disulfiram effect-impairing pharmaco-interactions with comedications are better understood, and functional delivery systems are available that target disulfiram specifically to the glioblastoma cells and that accumulate disulfiram and its active metabolites in the tumor above the concentrations reached by ingestion of maximally tolerable disulfiram doses. 

## Figures and Tables

**Figure 1 biomolecules-11-01561-f001:**
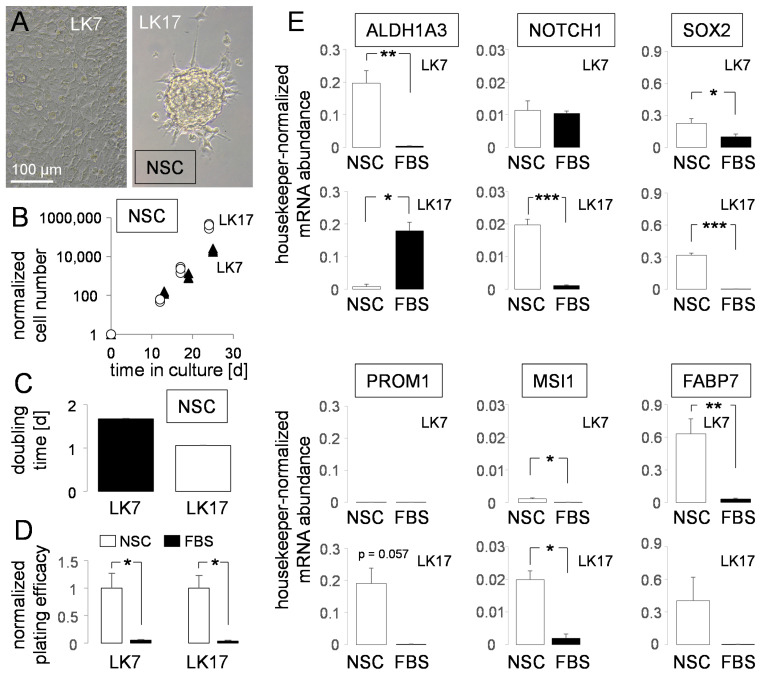
Stem-cell properties of LK7 and LK17 pGSCs. (**A**) Light micrographs showing the growth phenotype of LK7 (left) and LK17 (right) pGSCs. (**B**,**C**) Mean (±SE, *n* = 3) cell number (**A**) and doubling time (**B**) of LK7 (closed symbols/bar) and LK17 (open symbols/bar) cells during exponential growth in NSC medium. (**D**) Mean (±SE, *n* = 4) normalized plating efficiencies as a measure of clonogenicity of LK7 (left) and LK17 (right) pGSCs grown in NSC (open bars) and tumor “bulk” cell-differentiating FBS-containing medium. (**E**) Mean (±SE, *n* = 3–8) housekeeper-normalized abundance of mRNAs encoding stemness markers (as indicated) in LK7 (1st and 3rd line) and LK17 pGSCs (2nd and 4th line) grown in stem-cell-enriching NSC medium (open bars) or upon FBS-mediated “differentiation” into “bulk” tumor cells in 10% FBS/RPMI-1640 medium (closed bars). *, ** and *** indicate *p* ≤ 0.05, 0.01 and 0.001, respectively, Welch-corrected two-tailed *t*-test.

**Figure 2 biomolecules-11-01561-f002:**
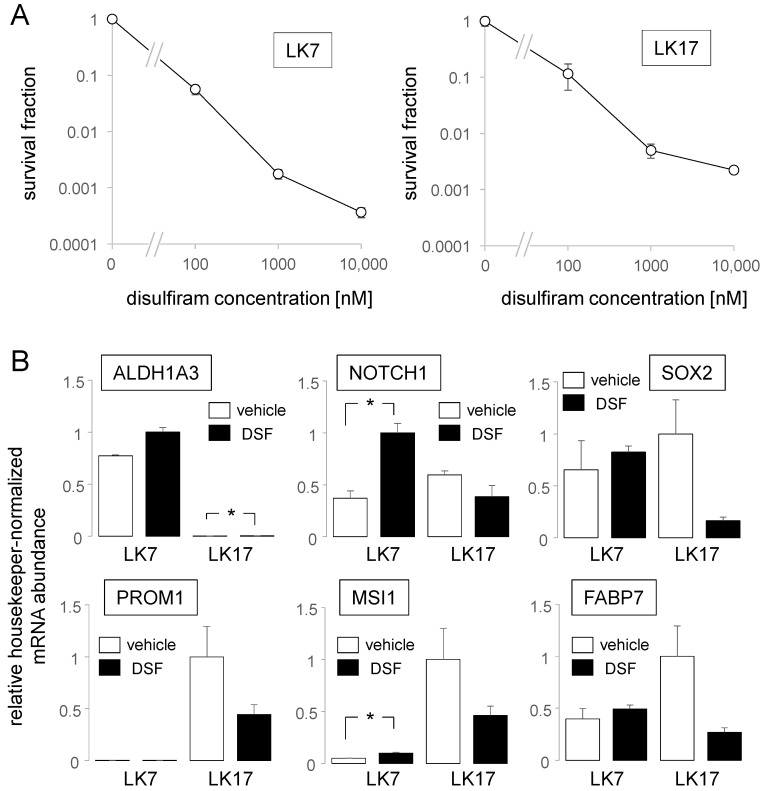
Disulfiram/Cu^2+^ inhibits clonogenic survival and modulates stem-cell properties of LK7 and LK17 pGSCs. (**A**) Relationship between mean survival fraction (±SE, *n* = 4–12) and the disulfiram (DSF) concentration of LK7 (left) and LK17 pGSCs (right) after cotreatment with disulfiram (0–10.000 nM) and CuSO_4_ (100 nM). Survival fractions were recorded in NSC medium by limited dilution assay. Absolute plating efficiencies at 0 nM disulfiram were 0.83 in LK7 and 0.11 in LK17 pGSCs. (**B**) Mean (±SE, *n* = 3) relative housekeeper-normalized abundance of mRNAs encoding stemness markers (as indicated) in LK7 (left) and LK17 cells (right) grown either in vehicle- (open bars) or DSF-containing NSC medium (closed bars). * indicates *p* ≤ 0.05, Welch-corrected two-tailed *t*-test.

**Figure 3 biomolecules-11-01561-f003:**
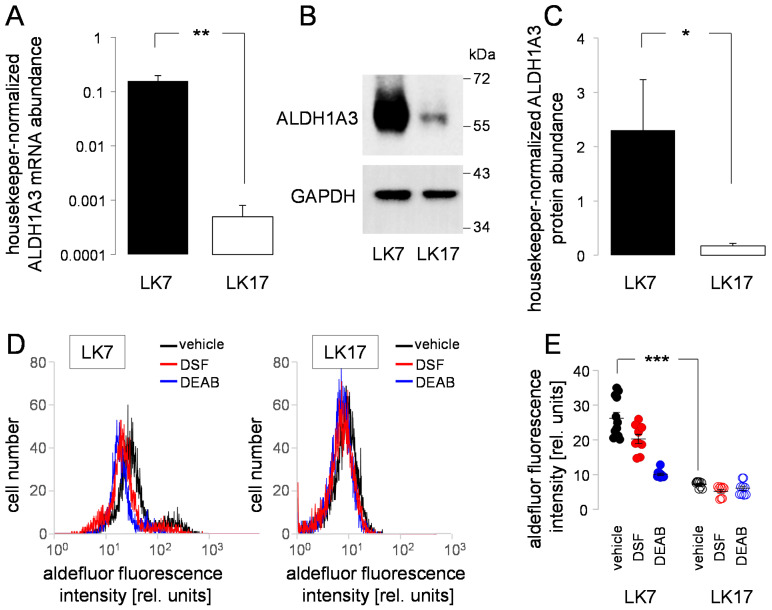
Primary glioblastoma stem-cell cultures LK7 and LK17 differ in ALDH1A3 mRNA and protein abundance and in ALDH activity. (**A**) Mean (±SE, *n* = 7–9) housekeeper-normalized *ALDH1A3* mRNA abundance of LK7 (left) and LK17 cells (right) as quantified by real-time RT-PCR. (**B**) Representative immunoblots of total lysates from LK7 (left) and LK17 (right) cells probed against ALDH1A3 (top) and—for loading control—GAPDH (bottom). (**C**) Mean (±SE, *n* = 9–10) housekeeper-normalized ALDH1A3 protein abundance of LK7 (left) and LK17 cells (right) determined as in (**B**) by immunoblotting. (**D**) Representative histograms recorded by flow cytometry showing the aldefluor-specific fluorescence intensity of LK7 (left) and LK17 (right) cells after incubation in the absence (vehicle, black) and presence of the ALDH inhibitor diethylaminobenzaldehyde (DEAB, 3 µM, blue) or disulfiram (DSF, 100 nM, red). (**E**) Individual and mean (±SE, *n* = 9–12) aldefluor fluorescence intensities (geometrical means) measured as in (D) by flow cytometry in LK7 (left) and LK17 (right) cells after incubation with vehicle (black), disulfiram (red), or DEAB (blue). * and ** in (**A**,**C**) and *** in (**E**) indicate *p* ≤ 0.05, 0.01, and 0.001, respectively, as calculated by Welch-corrected two-tailed *t*-test (**A**,**C**) and nonparametric Kruskal–Wallis and Dunn’s multiple comparisons test (**E**).

**Figure 4 biomolecules-11-01561-f004:**
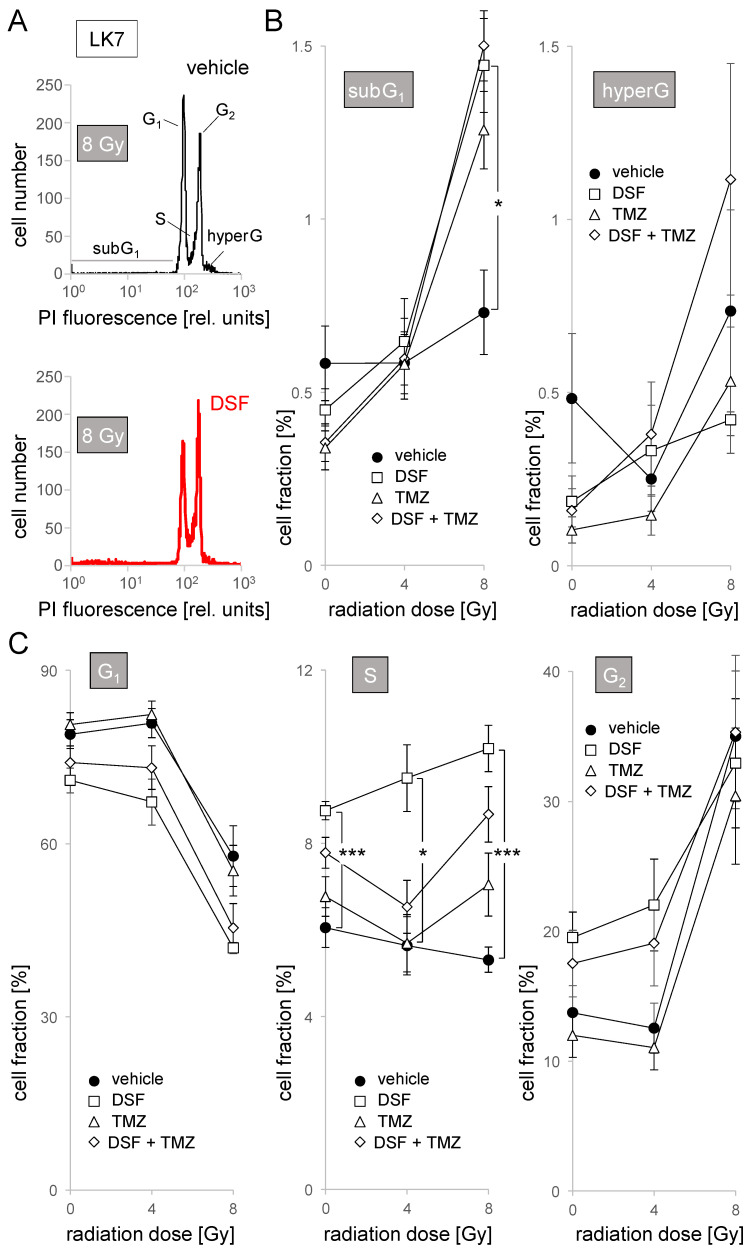
Disulfiram increases S population in LK7 cells. (**A**) Representative flow cytometry histograms showing the distribution of the DNA-specific propidium iodide (PI) fluorescence among vehicle (black) or DSF (100 nM, red) cotreated LK7 cells 48 h after irradiation with 8 Gy. (**B**,**C**) Dependence of the mean (±SE, *n* = 9) percentage of dead cells (subG_1_ population, (**B**), left), and of cells with polyploidy/polynuclearity (hyperG population, (**B**), right), as well as of cells in G_1_ ((**C**), left), S ((**C**), middle) and G_2_ ((**C**), right) phase of cell cycle on radiation dose and cotreatment with vehicle (closed circles), disulfiram (DSF, 100 nM, open squares), temozolomide (TMZ, 30 µM, open triangles) and DSF + TMZ (open diamonds). * and *** indicate *p* ≤ 0.05 and 0.001, respectively as calculated by Kruskal–Wallis nonparametric ANOVA and Dunn’s multiple comparisons test.

**Figure 5 biomolecules-11-01561-f005:**
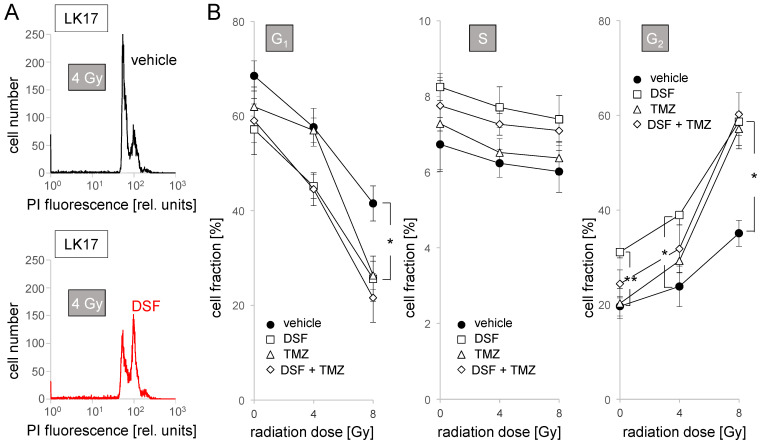
Disulfiram decreases G_1_ and increases G_2_ population in LK17 cells. (**A**) Representative flow cytometry histograms showing the distribution of the DNA-specific propidium iodide (PI) fluorescence among vehicle (black) or DSF (100 nM, red) cotreated LK17 cells 48 h after irradiation with 4 Gy. (**B**) Dependence of the mean (±SE, *n* = 6–9) percentage of cells in G_1_ (left), S (middle), and G_2_ (right) phase of cell cycle on radiation dose and cotreatment with vehicle (closed circles), disulfiram (DSF, 100 nM, open squares), temozolomide (TMZ, 30 µM, open triangles) and DSF + TMZ (open diamonds). * and ** indicate *p* ≤ 0.05 and 0.01, respectively as calculated by Kruskal–Wallis nonparametric ANOVA and Dunn’s multiple comparisons test.

**Figure 6 biomolecules-11-01561-f006:**
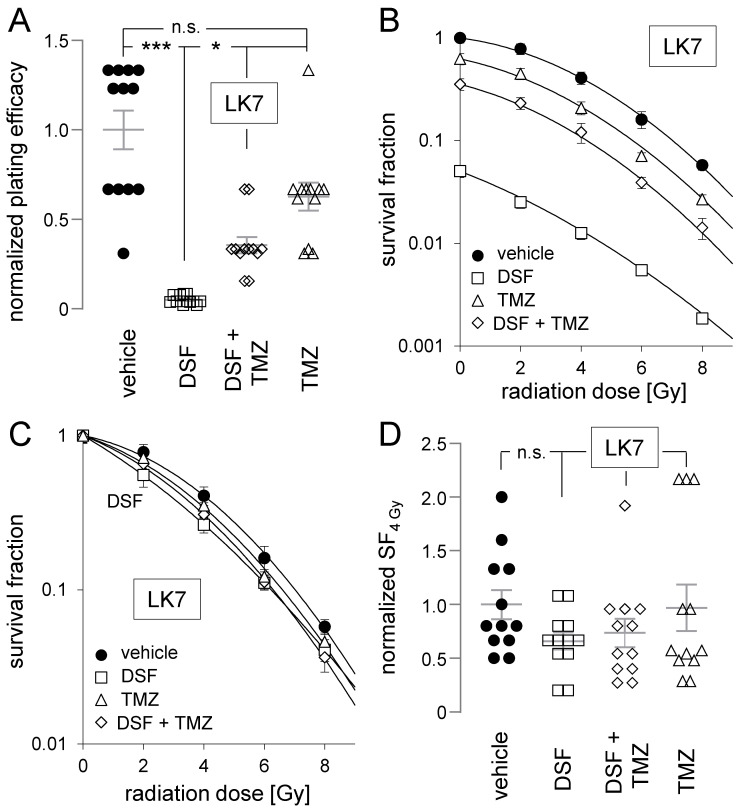
Disulfiram impairs clonogenic survival of LK7 cells. Neither disulfiram nor temozolomide radiosensitizes LK7 cells, and temozolomide attenuates the clonogenic survival-diminishing effect of disulfiram. (**A**–**D**) Dependence of normalized plating efficiency (**A**), vehicle-0 Gy control-normalized survival fraction (**B**), experimental arm-specific 0 Gy control-normalized survival fraction (**C**), and survival fraction after irradiation with 4 Gy (SF_4 Gy_, (**D**)) of LK7 cells on radiation dose (in (**B**,**C**)) and on cotreatment (in (**A**–**D**)) with vehicle (closed circles), disulfiram (DSF, 100 nM, open squares), temozolomide (TMZ, 30 µM, open triangles) and DSF + TMZ (open diamonds). Shown are means (±SE, *n* = 12) and additionally individual values in (**A**,**D**). Mean absolute (±SE, *n* = 12) plating efficiency of 0 Gy/vehicle-treated LK7 cells was 0.77 ± 0.29. *, ***, and n.s. indicate *p* ≤ 0.05, *p* ≤ 0.01, and not significantly differing, respectively, as calculated by nonparametric Kruskal–Wallis with Dunn’s multiple comparison test. To maintain clarity, significance asterisks were not sketched in (**B**).

**Figure 7 biomolecules-11-01561-f007:**
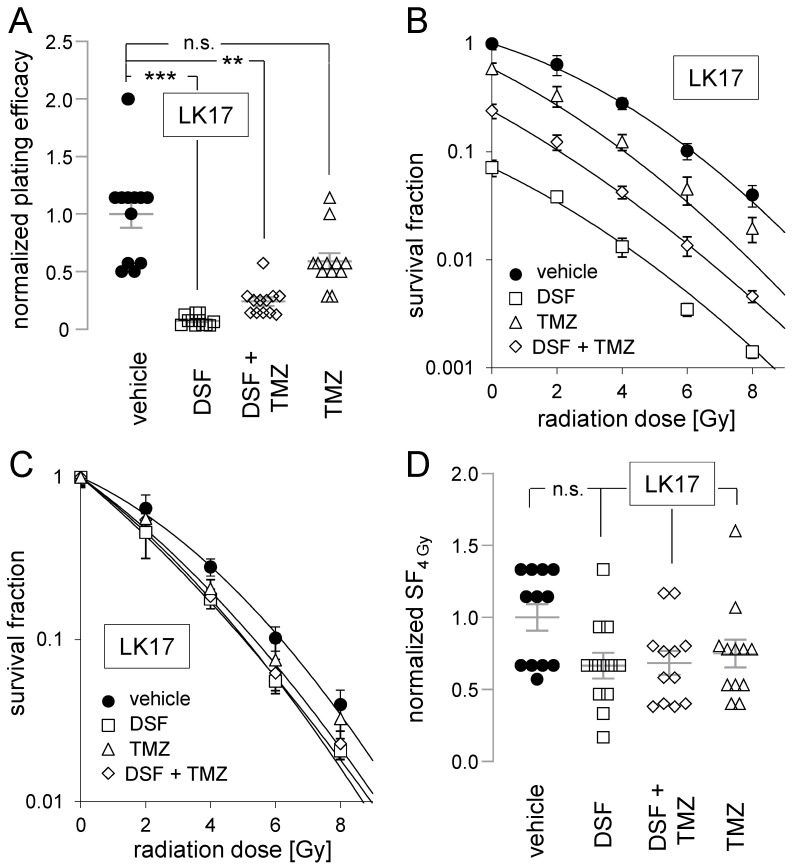
Disulfiram impairs clonogenic survival of LK17 cells. Neither disulfiram nor temozolomide radiosensitizes LK17 cells. (**A**–**D**). Dependence of normalized plating efficiency (**A**), vehicle-0 Gy control-normalized survival fraction (**B**), experimental arm-specific 0 Gy control-normalized survival fraction (**C**), and survival fraction after irradiation with 4 Gy (SF_4 Gy_, (**D**)) of LK17 cells on radiation dose (in (**B**,**C**)) and on cotreatment (in (**A**–**D**)) with vehicle (closed circles), disulfiram (DSF, 100 nM, open squares), temozolomide (TMZ, 30 µM, open triangles) and DSF + TMZ (open diamonds). Shown are means (±SE, *n* = 12) and additionally individual values in (**A**,**D**). Mean (±SE, *n* = 12) absolute plating efficiency of 0 Gy/vehicle-treated LK17 cells was 0.75 ± 0.09. **, *** and n.s. indicate *p* ≤ 0.01, *p* ≤ 0.001, and not significantly differing, respectively, as calculated by nonparametric Kruskal–Wallis with Dunn’s multiple comparison test.

## Data Availability

The datasets analyzed during the current study are available from the corresponding author upon reasonable request.

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
