# Peer review of "Repurposing Disulfiram for Targeting of Glioblastoma Stem Cells: An In Vitro Study"

_biomolecules, 2021, doi:10.3390/biom11111561_

Round 1

Reviewer 1 Report

Regarding the clonogenicity of the used glioblastoma cell lines in the presence or absence of disulfiram and/or temozolomide, the authors speculate that temozolomide reverses the strong inhibition of clonogenicity by 
disulfiram by an interference of temozolomide with disulfiram-triggered pathways that impair clongenic survival, but that the underlying 
cellular/molecular mechanisms are only of academic interest and do not bear any consequences for the clinical setting. I disagree with this opinion, because the authors' finding that the survival fraction of DSF-treated cells is increased ~10-fold when these cells are cotreated with TMZ suggests that the mechanism by which DSF induces cell killing can be subject to significant pharmaco-interactions. The authors suggest that disulfiram might be tested in future strategies for second line chemotherapy in glioblastoma if functional delivery systems are available, but without proper understanding of the underlying mechanism of this DSF-TMZ interaction, the success of clinical trials with disulfiram beyond temozolomide progression would be jeopardized because we do not know whether or not other comedications also lead to this interaction and thus negate the effects of DSF. I can therefore only give the present paper an "average" rating with respect to the interest to the readers.

Author Response

Dear Reviewer,

Thank your for your comments. According to your concerns, we deleted the sentence "Therefore, the molecular characterization of the observed disulfiram-antagonizing temozolomide effect would not have much clinical impact and, hence, was not addressed by the present study. " from our discussion section (lines 503-505) and adopted your argumention by adding the following paragraph at this text passage "

In addition, this observation suggests that the tumoricidal effect of disulfiram may be sensitive to pharmaco-interactions with co-medications. The understanding of such pharmaco-interactions, however, is a prerequisite for the success of future clinical trials using disulfiram for second line therapy in glioblastoma patients with tumor progression during temozolomide maintenance therapy. The analysis of the molecular mechanism of such pharmaco-interactions (here, the temozolomide-disulfiram interaction), however, goes beyond the scope of the present study".

Furthermore, we extended the concluding remarks accordingly and wrote in the revised manuscript:

"Disulfiram/Cu2+ might be tested in future strategies for second line chemotherapy in glioblastoma if disulfiram effect-impairing pharmaco-interactions with co-medications are better understood and functional delivery systems are available that target disulfiram specifically to the glioblastoma cells and that accumulate disulfiram and its active metabolites in the tumor above the concentrations actually reached by ingestion of maximally tolerable disulfiram doses."

With best regards,

Stephan Huber

Reviewer 2 Report

The resubmitted manuscript by Zirjacks et al. has been significantly revised and is now suitable for publication in Biomolecules. All my comments from the previous review have been incorporated. I have a few more editorial comments. The authors should trace the entire manuscript for the notation of gene and protein names. The names of genes used should be written in italics, while proteins should be written normally or in capital letters according to the accepted notation in databases.
Line 279-280 remove the space

Author Response

Dear Reviewer,

Thank you, indeed, for your valuable suggestions and comments which have made our aims, our argumentation, and the novelity of our data more comprehensible. According to your suggestions, we have traced the entire manuscript for the notation of gene and protein names and removed the space  between line 279-280. 

With best regards,

Stephan Huber

Reviewer 3 Report

The authors answered my questions and revised their manuscript accordingly.

Author Response

Dear Reviewer,

Thank you, indeed, for your valuable suggestions and comments which have made our aims, our argumentationand, and the novelity of our data more comprehensible.

With best regards,

Stephan Huber